# Tumor-Associated Macrophages in Bladder Cancer: Biological Role, Impact on Therapeutic Response and Perspectives for Immunotherapy

**DOI:** 10.3390/cancers13184712

**Published:** 2021-09-21

**Authors:** Marine M. Leblond, Hana Zdimerova, Emma Desponds, Grégory Verdeil

**Affiliations:** 1UNICAEN, CEA, CNRS, ISTCT/CERVOxy Group, GIP CYCERON, Normandie University, 14000 Caen, France; leblond@cyceron.fr; 2Department of Oncology UNIL CHUV, University of Lausanne, 1015 Lausanne, Switzerland; hana.zdimerova@unil.ch (H.Z.); emma.desponds@unil.ch (E.D.)

**Keywords:** macrophages, bladder cancer, macrophage-targeting immunotherapy

## Abstract

**Simple Summary:**

Tumor-associated macrophages (TAMs) play major roles in solid tumor development. They can have both anti-tumor and pro-tumor properties depending on their polarization. In this review, we summarize the observations and associations made between the presence of TAMs and their subtypes within bladder cancer and the type of disease, its evolution, the prognostic value for patients and the impact on current treatments. Only few studies focused on the effect of targeting TAMs in bladder cancer thus far. We propose several potential targets/treatments that may benefit for the limitation of pro-tumor TAMs and thus for the improvement of bladder cancer therapies.

**Abstract:**

Tumor-associated macrophages (TAMs) are one of the most abundant infiltrating immune cells of solid tumors. Despite their possible dual role, i.e., pro- or anti-tumoral, there is considerable evidence showing that the accumulation of TAMs promotes tumor progression rather than slowing it. Several strategies are being developed and clinically tested to target these cells. Bladder cancer (BCa) is one of the most common cancers, and despite heavy treatments, including immune checkpoint inhibitors (ICIs), the overall patient survival for advanced BCa is still poor. TAMs are present in bladder tumors and play a significant role in BCa development. However, few investigations have analyzed the effect of targeting TAMs in BCa. In this review, we focus on the importance of TAMs in a cancerous bladder, their association with patient outcome and treatment efficiency as well as on how current BCa treatments impact these cells. We also report different strategies used in other cancer types to develop new immunotherapeutic strategies with the aim of improving BCa management through TAMs targeting.

## 1. Introduction

### 1.1. Tumor-Associated Macrophages

Macrophages are phagocytic immune cells found in most tissues, including healthy bladder [1], with diverse functionalities. They are essential in maintaining homeostasis through their sentinel functions and ability to adapt and respond to physiological changes or challenges from the outside. They maintain tissue integrity by eliminating damaged cells and matrices and play a role during development, regulating tissue remodeling. Moreover, they are important players in host defense and partake in the immune response.

In the mouse bladder, a large population of macrophages is present in the submucosa of the bladder and increases upon infection [2]. Their activation through pattern recognition receptors (PRRs) and intracellular receptors, such as the inflammasomes, triggers cytokine and chemokine production [3]. Macrophages have been shown to negatively affect the development of adaptive immunity to urinary tract infection [1,4] and although the functions of macrophages in bladder immunity, tissue integrity and homeostasis are not well understood, there is mounting evidence that resident macrophages can have a negative impact in bladder disease.

In cancer, macrophages are one of the major populations of infiltrating leukocytes in solid tumors [5]. Tissue-resident macrophages or monocyte-derived macrophages can infiltrate the tumor and are then called tumor-associated macrophages (TAMs). In the tumor, these cells can present two extreme phenotypes across a continuum of activation states, which are known as M1-like (pro-inflammatory) and M2-like (anti-inflammatory) TAMs [6]. M1-like TAMs have been reported to inhibit tumor development, progression, angiogenesis and promote adaptive immune responses through the secretion of pro-inflammatory cytokines [7]. M1-like TAMs can kill tumor cells through, for instance, nitric oxide (NO) production. They also have the capacity to present tumor antigen to Th1 CD4^+^ T cells and drive the activity of cytotoxic CD8^+^ T cells [8]. During tumor progression, tumor cells can subvert TAMs to prevent M1-like accumulation and favor M2-like pro-tumor TAMs. These macrophages stimulate tumor initiation, progression and survival. They promote tumor growth and angiogenesis by providing cell growth factors and angiogenic molecules [9]. M2-like TAMs also favor cancer metastasis, as they stimulate tumor cell motility, invasion, and extravasation. In addition, M2-like TAMs have the capacity to secrete anti-inflammatory cytokines or inhibitory molecules, such as programmed death-ligand 1 (PD-L1) [10]. As a result, the activity of cytotoxic CD8^+^ T cells is suppressed, preventing tumor cell elimination [11]. Likewise, CD4^+^ T cells are prompted to differentiate into regulatory T cells, known to contribute to immune response suppression [12]. 

To characterize macrophages, specific robust markers and markers to differentiate M1- from M2-like TAM are still subject of debate [13]. Macrophages have two origins. During embryonic development, the yolk sac gives rise to organ-resident macrophages which are locally self-maintained [14]. However, upon infection or tissue damage, macrophages are released from the bone marrow as immature monocytes that will circulate in the blood until they reach the targeted site. Two monocyte subsets can be described and present with different chemokine receptors and surface molecules: the inflammatory CD11 b^+^Ly6 C^hi^ (CD14^+^CD16^−^ in human) and the patrolling CD11 b^+^Ly6 C^low^ (CD14^+^CD16^+^ in human) monocytes [15]. Once monocytes enter the target tissue, they differentiate into macrophages expressing F4/80 and CD11 b in mice and CD14, CD68 and CD16 in humans [16,17]. Inducible nitric oxide synthase (iNOS) and arginase 1 (Arg1) can be used to describe M1- and M2-like TAM, respectively [18]. Other M2-like TAM markers were used, such as CD163, CD204, CD206, DC-SIGN, Galectin-9 or hypoxia-inducible factor-2α (HIF-2α). They were generally used alone in immunohistochemistry, making a comparison of possible different populations of M2-like macrophages impossible. We will further discuss these populations and their importance in the context of BCa. Within the same tumor, it has been shown that M1-like and M2-like TAMs can coexist. However, in advanced tumors, TAMs generally present an M2-like state that correlates with poor cancer prognosis [19].

A closely related subtype to macrophages which can also be recruited to the site of inflammation is the myeloid-derived suppressor cell (MDSC). MDSCs are known to be immature cells deriving from the myeloid lineage. These cells are a heterogeneous population, but in this review, we will focus on one subtype, the monocytic MDSCs (M-MDSCs), expressing CD11 b^+^Ly6 C^high^Ly6 G^−^ in mice and CD33^+^CD14^+^HLA-DR^low^CD15^−^ in human [12]. It is still unclear whether these cells give rise to macrophages or represent a specific terminal population [11]. The initial defining feature of MDSCs is their activity to suppress the adaptive immune response, thus, potentially influencing the fate of certain diseases, such as cancer. 

### 1.2. Bladder Cancer

Bladder cancer (BCa) represents the fourth most common cancer type in men [20], with about 430,000 new cases per year worldwide [21]. The most common forms of BCa are urothelial carcinomas, which are classified using the tumor–node–metastasis system (TNM system) that characterizes first the degree of invasion of the primary tumor (pathologic T (pT) stage from pTis to T4), then tumor spreading to nearby lymph nodes, and lastly, whether there is a development of distant metastases. These tumors are also graded according to the cellular anaplasia which predicts the low-risk (low grade) or the high-risk (high grade) of tumor growth, spreading or recurrence [22]. 

Among urothelial carcinomas, about 75% are a non-muscle-invasive bladder cancer (NMIBC, pTis-T1) where the tumor localizes to the urothelium or lamina propria but does not invade the muscle layers. The other 25% of urothelial carcinomas are a muscle-invasive bladder cancer (MIBC, pT2-T4) and metastatic disease [22]. Within 5 years, between 50% and 70% of NMIBC will recur and 10% to 30% will progress toward MIBC or metastatic disease despite treatments [21]. For MIBC, the 5-year survival is about 50% with 50% of patients developing metastases. Patients with metastatic disease have a median survival of 15 months [23]. These numbers highlight the urgent need to improve treatment management for BCa patients, but this first requires a detailed understanding of BCa pathogenesis. 

## 2. Macrophages in Bladder Cancer

### 2.1. Association of Macrophages, Clinicopathological Features and Outcomes in Bladder Cancer

The tumor immune microenvironment of BCa remains poorly investigated compared to other solid tumors. Macrophages represent the most abundant infiltrating immune cells with a mean density of 14.55 cells/mm^2^ in the tumor core of MIBC [24]. Several studies on patient samples have reported that TAMs, and specifically M2-like TAMs, are present in both the stroma and the tumor core of BCa [24,25,26,27]. 

In NMIBC, TAM infiltration is less intense in the bladder wall compared to normal control bladder wall [28]. Studies analyzing NMIBC and MIBC cohorts have reported that TAM count, determined by the pan-macrophage marker CD68, positively correlates with high pathologic T stages and a high grade [29,30,31,32,33]. However, two studies have declared no correlation between TAM count, detected by CD204 or CD163 markers, and tumor stage or grade [34,35]. One explanation for these opposite results may be the use of unique and different markers to detect TAMs in human tissues emphasizing the lack of robust macrophage markers for pathologists [13]. 

The CD163^+^ M2-like phenotype is also associated with tumor stage and grade in patients with BCa [26,36,37,38]. Wang and colleagues have reported that the localization of M2-like macrophages in the tumor tissue is important to consider as their density in the stroma, but not in the tumor core, is positively associated with tumor stage [25]. The correlation between TAMs, especially M2-like TAMs, and grade and stage of BCa has been confirmed at the largest scale by RNA-seq on the TCGA MIBC cohort [38]. In support of this, it was observed in a genetically-engineered mouse model of BCa that TAM number, composed of M2-like TAMs, increases with tumor progression [39]. However, TAM count is not associated with any other clinicopathological feature such as gender, age, tumor volume or multifocality in BCa [26,32,34]. 

In peripheral blood, CD33^+^ and CD14^+^ myeloid cells increase in BCa patients compared to healthy donors [32,40,41]. In parallel to TAMs, the level of circulating myeloid cells is higher in BCa patients with pT2-T4 stages than with pTa-T1 stages [32] suggesting a communication between the systemic and tumor inflammation. 

The increase of TAMs with malignant progression suggests a role of these cells in BCa aggressiveness and clinical outcome. The presence of TAMs has been correlated with several unfavorable clinical outcomes in different solid tumors, including BCa (Table 1). Additionally, TAMs are associated with tumor recurrence in NMIBC patients [42,43]. In both NMIBC and MIBC patients, a high number of TAMs is associated with a higher risk of tumor progression [31,44], as well as a worse progression-free survival (PFS) and overall survival (OS) [27,29,35,45]. Depending on the phenotype and localization, macrophages can be associated with opposite clinical outcomes in BCa. By using DC-SIGN and CD68 to characterize M2-like TAMs, it was shown that they may contribute to tumor progression and poor prognosis [46]. Study of CD204^+^ staining of macrophages showed that their presence in the tumor stroma, but not in the tumor core, is associated with poor OS [25]. Using a broader M2-like signature, TCGA database analyses revealed that M2-like TAMs are significantly associated with a decreased OS and disease-free survival (DFS) [38,47]. In the blood, patients with a high level of circulating myeloid cells have a poor OS [32]. Conversely, CD169^+^ M1-like macrophages in the tumor-draining lymph nodes [48], but not in the tumor [25], are positively correlated with a favorable prognosis in MIBC patients. However, single-cell RNA-seq from immune-infiltrating cells revealed the wide diversity of TAMs in MIBC patients [49]. In this study, six different subsets of monocytes/macrophages were found in MIBC tumor, and their gene signatures did not correlate with the classical M1-like versus M2-like signatures. Moreover, the authors also identified another subset of TAM that shares both TAM-like and MDSC-like gene signatures [49], supporting the theory of the MDSC-to-macrophage differentiation. In the future, determining the origin of TAM (from tissue-resident macrophages, monocytes-derived macrophages or MDSC differentiation) could improve knowledge on TAM diversity and complexity but this is still limited by the lack of specific markers for each subtype. Altogether, this shows how complex it is to classify these cells by using a limited number of markers, leading to possible false positive count [13]. 

In summary, despite possible caveats related to the limited number of markers used to characterize TAM, M2-like TAMs were associated with higher tumor grade and bad prognosis for patients. Understanding how BCa favors this accumulation becomes critical in finding new strategies to prevent this accumulation and to treat patients.

### 2.2. Bladder Cancer Recruits TAMs

Tumor cells are the first actors to influence macrophages recruitment, as they are known to produce several chemokines and cytokines. In several solid tumors, macrophages are recruited to the tumor site through a gradient of chemotactic molecules, such as CCL2 (also known as MCP-1), colony stimulating factor 1 (CSF-1, also known as M-CSF) or several CXCL chemokines. In BCa (Figure 1), tumor cells are implicated in macrophage recruitment via the secretion of CCL2 [51] and MIF/CXCL2 [27]. However, the tumor is heterogeneous and different tumor clones can be found in the same tumor mass and can affect macrophages differently. Cheah and colleagues indicated that CD14^high^ bladder tumor cells developed more vascularized and more infiltrated tumors than did CD14^low^ bladder tumor cells. This was explained by the fact that CD14^high^ tumor cells have a higher production of IL6, IL8/CXCL5, CXCL1, CXCL2, M-CSF, VEGF-A, FGF-2 than CD14^low^ tumor cells [52]. Tumor heterogeneity is also the result of the oxygen level in the tumor microenvironment where hypoxia is a common characteristic. In BCa, TAM infiltration is higher in tumors presenting a high expression of HIF-1α or HIF-2α [50,53] and they are particularly concentrated in the hypoxic/necrotic areas of the tumor core [54], suggesting that hypoxic cells synthetize molecules that recruit TAMs in these specific areas. These mechanisms of TAM recruitment in BCa are potential new targets as we will discuss later in this review.

### 2.3. Bladder Cancer Favors M2-like Polarization of TAM to Promote Tumor Progression

In addition to TAM recruitment, bladder tumors also influence TAM phenotype by inducing the establishment of an M2-like phenotype that will enhance tumor development in return (Figure 1). Bladder tumor cells increase expression of CD206, CD163, PD-L1 and IL10 in macrophages [55,56] through the secretion of IL10 [57], chemokines [42,51], metabolic products [55,58], growth factors [37] and micro-RNA via exosome exchange [59]. Moreover, hypoxic areas, where TAMs are particularly concentrated in BCa [54], favor M2-like macrophages [60] through intracellular signaling and tumor-derived metabolites [61,62], which could contribute to the accumulation of pro-tumor macrophages in BCa.

In return, M2-like macrophages provide support for tumor progression. They contribute to an increase in bladder tumor cell proliferation and viability by the secretion of CXCL1 and collagen-I [63,64]. Macrophage count is also correlated with lymphatic metastasis underlying their role in the development of metastasis [29,35,65]. Macrophages promote lymphangiogenesis by the secretion of VEGF-C/D [51,66] and increase the ability of bladder tumor cells to form colonies and generate tumor spheres [56,67]. Macrophages further favor the development of metastasis by inducing tumor cell invasion through the production of CXCL8 and osteopontin [33,59,68,69]. This, together with the fact that in the tumor core TAMs are particularly concentrated at the proximity of basal-like layer/invasive front [37], indicates that macrophages could be involved in the transition from NMIBC to MIBC by supporting the invasion of tumor cells through the muscle layers. Moreover, because of the positive correlation between TAM counts and micro-vessel counts [42,63,70] and because of their localization in hypoxic/necrotic areas, TAMs are suspected to promote tumor angiogenesis. Macrophages provoke angiogenesis directly by producing VEGF [71] or indirectly by stimulating tumor cells to produce pro-angiogenic factors [67,68,72]. Finally, macrophages help to promote tumor progression by their capacity to inhibit anti-tumor immunity. Two studies have shown from fresh human tissues that CCR8^+^ TAMs and hyaluronidase 2-expressing TAMs increase the level of inflammatory factors in bladder tumor tissues, such as IL6, CXCL8 and CCL2 [71,72]. In parallel, CCR8^−^ TAMs secrete CCL1 that will activate CCR8 on nearby TAMs increasing tumor inflammation [71]. In this study, Eruslanov and coworkers stated that CCR8^+^ TAMs are able to induce FoxP3 in lymphocytes, thus favoring regulatory T cells (Tregs) in the tumor microenvironment of BCa. The accumulation of Tregs in the bladder tissue is promoted by DC-SIGN^+^ TAMs [46]. The accumulation of suppressor lymphocytes is accompanied by a reduction of cytotoxic lymphocytes coordinated by DC-SIGN^+^ TAMs and PD-L1-expressing myeloid cells [46,55,57]. Furthermore, the expression of Galectin-9 by TAMs seems to promote exhaustion in T cells as they are associated with a decrease of IFNγ, GrzB, PRF1 and an increase of PD-1 and Tim3 in CD45^+^ cells [73]. Suppression of myeloid cells leads to an increased infiltration of CD8^+^ T cells in the tumor [74]. It seems that several specific subsets of M2-like macrophages are present in the environment of BCa, each having a specific role in tumor development. This concept shows the difficulty in finding common markers that can target all M2-like macrophages without depleting M1-like macrophages. As detailed later, M1-like macrophages can be important for the anti-tumor response during BCa therapy.

## 3. Macrophages Influence Bladder Cancer Treatments

Currently, the major challenge in the management of BCa is the prevention of recurrence and progression of the disease. For NMIBC, the initial step is the transurethral resection of the bladder tumor (TURBT). For patients with intermediate- or high-risk disease, the gold-standard treatment is TURBT followed by intravesical immunotherapy based on Bacillus Calmette-Guérin (BCG). However, BCG failures can occur and for these patients the most effective treatment is radical cystectomy (RC) [21]. For MIBC patients, the gold-standard treatment consists of RC with or without neoadjuvant systemic cisplatin-based chemotherapy. Patients with metastatic urothelial carcinoma are treated with adjuvant chemotherapy after RC [21]. For advanced MIBC, immune checkpoint inhibitors (ICIs) are now approved as second-line therapy and as first-line treatment for cisplatin-ineligible patients [75].

Poor patient survival highlights the fact that BCa tumors are resistant. This can rely on the interaction between tumor cells and their surroundings. Among them, macrophages were described to be involved in several types of tumor-treatment resistance [76], including in BCa (Table 2).

### 3.1. Macrophages and NMIBC Treatments

#### 3.1.1. Transurethral Resection of Bladder Tumor (TURBT) 

The role of macrophages on TURBT efficacy is poorly documented, mainly because there are few patients that undergo tumor removal only, generating powerless cohorts for statistics. Nevertheless, tumors removed via TURBT present lower TAM counts than tumors undergoing RC [31] and a high TAM number is associated with shorter survival in TURBT patients [31]. In a cohort that underwent only TURBT or RC, Wang and colleagues showed that total macrophage, and CD204^+^ macrophage, only in the stroma and not in the tumor core were associated with poor OS after surgery [25]. 

#### 3.1.2. Bacillus Calmette-Guérin (BCG) 

As reviewed in detail by Redelman-Sidi and colleagues, the therapeutic effect of BCG is based on the induction of a local inflammation that includes cytokine/chemokine secretion and the recruitment, as well as the activation, of several immune cells including macrophages [94]. BCG treatment is known to increase monocytes in the blood [95], as well as macrophage infiltration in the bladder wall [28] and urine [41,96]. These macrophages are probably recruited to the bladder from monocytes through chemokine production, as it is known that BCG treatment increases the level of chemokines, such as MCP-1 and CXCL8, in the serum and urine of patients [97,98,99,100]. Chemokines can be secreted by healthy and tumor urothelial cells, as well as PBMCs after BCG exposure [99]. Several cytokines (IL1, IL6, TNFα and IFNγ) were also increased in the urine of patients after BCG treatment [97,101], suggesting the activation of macrophages after the intravesical administration of BCG. In vitro experiments confirmed that BCG treatment efficiently induces the production of Th1-cytokines in macrophages but also macrophage-mediated cytotoxicity toward bladder tumor cells [102,103,104]. Consequently, macrophages are important in the beneficial response to BCG immunotherapy. This was supported by Kitamura and coworkers, who have communicated that a higher CD68^+^ cell count in the tumor following intravesical BCG immunotherapy was correlated with better RFS in NMIBC patients [78].

Despite their importance in BCG treatment, high counts of TAMs pre-BCG treatment are associated with poor RFS [30,79,80,81] and PFS [34]. BCG can also induce pro-tumor functions in macrophages. Macrophage-produced IL10 following BCG exposure reduced the cytotoxic activity of macrophages themselves [105], as well as induced the expression of PD-L1 on antigen-presenting cells, including macrophages [106]. Increased expressions of PD-L1 together with PD-L2 and PD-1 were also observed on cells from patients’ urine after BCG treatment [95]. Moreover, BCG-stimulated macrophages can support the proliferation, differentiation and activation of fibroblasts [107], which are well-known to promote tumor progression [108]. Another hypothesis to explain this dual role of macrophages during the BCG response could be that pre-BCG treatment TAMs, which are mainly M2-like TAMs, could not be repolarized by BCG immunotherapy and that their phenotype would determine the outcomes of the BCG response. High counts of M2-like TAMs before BCG immunotherapy are correlated with BCG failure, whereas pre-BCG treatment M1-like TAMs are associated with better DFS [26,82,83]. Moreover, a low T cell to immunosuppressive myeloid cell ratio in the post-BCG urine, but presumably already present before BCG treatment, correlates with poor RFS and PFS [41]. Therefore, the beneficial effects of macrophages in BCG immunotherapy would be based on the recruitment and differentiation of fresh macrophages that will not be corrupted by a prolonged exposure to the tumor microenvironment. Furthermore, it seems that an immune signature, with a focus on macrophages, in tumor and/or urine of NMIBC patients before BCG immunotherapy can be a useful biomarker to discriminate BCG responders.

### 3.2. Macrophages and MIBC Treatments

#### 3.2.1. Radical cystectomy (RC)

As for TURBT, the impact of macrophages on RC is not well understood because of the small size of cohorts that will undergo only surgery. However, studies revealed that higher TAM counts were associated with higher rates of RC [29] and that tumors undergoing RC present higher TAM counts than tumors undergoing TURBT as mentioned before [31]. Total TAM number was not associated with survival [54] but specific subsets of TAMs were correlated with poor survival, such as MAC387^+^ TAMs [31] or HIF-2α^+^ TAMs [54] in RC patients. Moreover, it seems that circulating CD33^+^ cells could be predictive of a poor RC response as patients with low circulating CD33^+^ cells before RC experience a pathological complete response [77]. However, in this study, all RC patients were incorporated in the analysis, regardless of neoadjuvant or adjuvant therapies, and thus the predictive role of CD33^+^ PBMCs on surgery efficacy needs further investigation. 

Altogether, these studies support the fact that macrophages may be involved in bladder tumor recurrence and/or progression after tumor resection. One of the physiological functions of macrophages is their capacity to repair and remodel tissue after injury. As surgery inflicts damage to the affected tissue, one could hypothesize the likely concept that the macrophages in the surroundings/around the wound start the repair and in the process benefit remaining tumor cells.

#### 3.2.2. Chemotherapy

The effect of macrophages on chemotherapy in BCa patients depends on the subtype of macrophage and the disease stage. Taubert et al. declared that high TAM count was associated with poor outcomes in pT2-pT4 patients [84] contrary to Fu et al., who showed that an immune signature containing high TAM was correlated with increased OS and DFS in pT3-pT4 patients [24]. One explanation for the differences observed in the two studies could be that the CD68 marker is a pan-macrophage marker, but it cannot differentiate the specific subtypes. High infiltration of M2-like TAMs, such as CD163^+^ TAMs [84] and DC-SIGN^+^ TAMs [46], was associated with unresponsiveness to chemotherapy. However, including if galectin-9^+^ TAMs are associated with poor outcomes in BCa, this subset presents a survival benefit after adjuvant chemotherapy [73]. Circulating CD33^+^ HLA-DR^−^ cells were negatively associated with a pathologic complete response after neoadjuvant chemotherapy [85]. In summary, it seems that macrophages are detrimental for chemotherapy responses. 

#### 3.2.3. Immune Checkpoint Inhibitors (ICIs)

The use of PD-1 and PD-L1 blockade in BCa therapy is consistent with the fact that PD-1 and PD-L1 expression is detected in bladder tumors and that their expression increases with tumor stage and grade [109,110,111]. In the majority of RC patients, PD-1 was observed in the tumor area but not in the normal urothelium [112]. PD-1 can be expressed by tumor-infiltrating lymphocytes (TILs) [111,113] and TAMs. PD-1^+^ TAMs were found in 47.5% of MIBC patients and were positively associated to pT stage clinicopathological features [47]. PD-1^+^ TILs were also positively associated with pathological stage [114] and negatively associated with OS in BCa patients [111]. Concerning PD-L1, it can be expressed by both TAMs and bladder tumor cells [57,115]. Most studies have declared that PD-L1 expression, on infiltrating immune cells and bladder tumor cells, is correlated with a poor patient prognosis [47,111,113,116,117], except in pT1 NMIBC [118,119]. 

Despite the infatuation for ICIs and the fact that bladder tumors are one of the most immunogenic tumors [120], only 15–20% of patients will respond to PD-1/PD-L1 blockade [121]. Regarding the mechanisms behind resistance, Mariathasan and collaborators have demonstrated that the lack of response to Atezolizumab (anti-PD-L1) in metastatic BCa patients was associated with TGFβ-inducing cytotoxic T cell exclusion. TGFβ blockade restored the infiltration of T cells, inducing a profound anti-tumor immunity after anti-PD-L1 treatment [86]. Whether in this study, the authors declared that fibroblasts were responsible for the TGFβ-induced T cell exclusion, Peranzoni and colleagues have demonstrated that TAMs can also participate in T cell exclusion in lung cancer [122]. TAM depletion resulted in the increase of T cells in the tumor, which enhanced the efficacy of anti-PD-1 treatment in different tumor models [122,123]. We have shown that M1-like TAMs are important players for the efficacy of an anti-CD40/anti-PD-1 combo therapy in a MIBC mouse model [39]. This was also confirmed by other studies which demonstrated that a reeducation of M2-like TAM toward M1-like TAM enhanced the efficacy of anti-PD1 and anti-PD-L1 immunotherapies [124,125]. The M1-like TAM signature was associated with an improved prognosis after PD-1/PD-L1 blockade [126]. Thus, targeting of TAM should be directed toward M2-like macrophages.

Studies to determine biomarkers to predict anti-PD-1 and anti-PD-L1 responders focused mostly on PD-L1 expression. In phase II and III clinical trials investigating the safety and efficacy of Atezolizumab in BCa patients, high PD-L1 expression on immune cells, but not on tumor cells, was associated with response [86,87,88,89]. Other clinical trials evaluating Pembrolizumab (anti-PD-1) and Nivolumab (anti-PD-1)+ Ipilimumab (anti-CTLA4) efficiencies have declared that responses were observed regardless of PD-L1 expression levels [90,91]. A meta-analysis of 45 FDA-approved drugs indicated that PD-L1 expression was not predictive in 53.3% of cases across 15 tumor types [127]. Thus far, the expression of PD-L1 in BCa as a biomarker for the response to anti-PD-1/anti-PD-L1 immunotherapies is still controversial. Recent studies have analyzed if myeloid cells could be a relevant biomarker for ICI efficiency in BCa patients. Zeng et al. have analyzed the gene expression in the IMvigor210 cohort (Atezolizumab in BCa patients) and they declared that the M1-like TAM signature was a robust biomarker for predicting the prognosis and response to anti-PD-L1 [92]. Wang and collaborators demonstrated that a pro-tumorigenic inflammation signature was correlated with poor survival in the IMvigor210 and CheckMate275 (Nivolumab in BCa patients) cohorts [49]. They determine a score where TAMs and monocytes were not defined by the classical M1/M2 signature. With this scoring, they demonstrated that monocytes with a low score were enriched in the peripheral blood of metastatic BCa patients with resistance to anti-PD-L1 [49], suggesting that pro-tumorigenic myeloid cells could be a pertinent biomarker for ICI efficiency.

To conclude, it seems that pro-tumoral TAMs can limit the efficacy of ICIs from the PD-1/PD-L1 axis and thus may be relevant to discriminate responder from nonresponder patients. Because of their implication in treatment efficacy, the combination of current treatments with macrophage-targeting strategies is appealing in the case of BCa. However, it is important to consider that M1-like TAMs are beneficial for immunotherapies and that the drastic depletion of these cells is probably not the best strategy. 

## 4. Targeting TAMs to Improve Bladder Cancer Outcome

At the moment, BCG and PD-1/PD-L1 blocking antibodies are the only immunotherapeutic strategies for the management of BCa. However, considering the effect of TAMs on BCa progression and therapeutic efficacy, targeting them as future immunotherapeutic strategies to improve current treatments seems appropriate. Although several strategies to target TAMs are being clinically tested in different cancers [93] and include hundreds of registered clinical trials, only a few focus on BCa (Table 3). Here, we review several macrophage-targeting strategies that seem relevant for BCa (Figure 2).

### 4.1. Strategies to Inhibit Macrophage Recruitment in Bladder Cancer

In several solid tumors, inhibition of macrophage recruitment is one of the most tested strategies toward TAMs. As described above, strong evidence, such as the increase of myeloid cells in the blood parallel to bladder tumor progression [32], indicates that macrophages are recruited during systemic inflammation through tumor-derived chemokines and then accumulated in the bladder tumor. The tumor-derived chemokines implicated in TAM recruitment in BCa include CCL2, SDF-1 and the ligands of CXCR2.

#### 4.1.1. CCL2-CCR2 Axis

CCL2 is seen as a major chemokine involved in the recruitment of TAMs through the CCL2-CCR2 axis [128]. Several pre-clinical studies have demonstrated that targeting TAMs via the CCL2-CCR2 pathway, with CCR2 antagonists or CCL2 neutralizing antibodies, increases the survival of tumor-bearing mice [129,130]. Recruitment inhibition of myeloid cells through this axis improved chemotherapy [131], radiotherapy [132] and ICI [133] responses in different tumor models. However, Bonapace and colleagues have demonstrated that a discontinuation of anti-CCL2 treatment accelerates tumor development in a breast cancer model [134]. Concerning BCa, CCL2 was detected in the urine of patients and an increased level was correlated with an increased tumor stage and grade [135]. High CCL2 level also correlated with high CD68 and CD163 staining in tumor tissues, supporting the role of CCL2 in TAM recruitment [136]. It was shown that CCL2 can be produced by both tumor and myeloid cells [51,136,137], inducing an autocrine loop in the recruitment of TAMs. In BCa models, inhibition of TAMs with an anti-CCL2 neutralizing antibody results in a reduction of lymph node metastasis [51] and increased mouse survival after chemotherapy [138]. 

#### 4.1.2. Stromal Cell-Derived Factor 1 (SDF-1)-CXCR4 Axis

Another chemokine involved in TAM recruitment is SDF-1, also known as CXCL12, which binds to its receptor CXCR4 [128]. This axis was demonstrated to be involved in TAM recruitment in several solid tumors, especially to promote angiogenesis in hypoxic tumors [139]. TAM inhibition with a CXCR4 antagonist, AMD3100, delayed recurrence after radiotherapy [139,140] and chemotherapy [141] but also improved ICI efficiency by increasing T cell recruitment [142]. Blocking this axis seems to be a relevant strategy for BCa treatment as SDF-1 and CXCR4 are expressed in bladder tumor tissue [143]. Their expression is associated with tumor pT stage and grade [144,145,146] but also with poor patient OS [147]. Moreover, TCGA analysis demonstrated that SDF-1 gene expression positively correlates with M2-like TAMs [146], confirming the relationship between the SDF-1-CXCR4 axis and TAM recruitment in BCa. Despite that, no preclinical study has evaluated the effect of blocking TAMs via the SDF-1-CXCR4 pathway in BCa. 

#### 4.1.3. CXCL-CXCR2 Axis

CXCR2 is a C-X-C chemokine receptor that is known to recruit immunosuppressive myeloid cells through their secretion of several chemokines, such as CXCL1, CXCL2, CXCL5 and CXCL8/IL8 [148]. While most studies focused on the neutrophil/polymorphonuclear-MDSC recruitment via CXCR2, CXCL5-CXCR2 axis blocking resulted in a dramatic reduction of monocytic myeloid cells in prostate tumors [149]. In BCa, Zhang et al. demonstrated that CXCL2, produced by BCa cells, was responsible for the recruitment of CXCR2-expressing CD33^+^ myeloid cells. Also, a high CXCL2 expression was correlated with poor OS [27]. Moreover, CXCL1 levels in the urine of BCa patients were higher compared to control subjects but no statistical differences were noted between high and low grade or tumor stages [150,151]. However, in bladder tumor tissues, CXCL1 staining was observed in 40% of pTa tumors as well as 75% of pT1-4 tumors [152] and its expression was increased with tumor stage and grade [153]. Increased levels of CXCL5 in bladder tumors, as well as in urine, were associated with tumor stage, grade and lymph node metastasis [154,155]. Concerning CXCL8, a higher level was detected in the urine of patients compared to noncancer patients [156] and a stronger CXCL8 expression was observed in high grade, compared to low grade, bladder tumors [68]. High expression of CXCL8 in tumor tissues was correlated with a high infiltration of TAMs [68]. Moreover, BCG immunotherapy is known to induce CXCL8 production which is essential for the recruitment of several immune cells [94], supporting the hypothesis that the CXCR2-CXCL8 axis can be involved in TAM recruitment in BCa. To the best of our knowledge, only one preclinical study has demonstrated that blocking CXCL8 with a blocking antibody resulted in decreased tumor growth and invasion in an athymic mouse model [157]. 

Targeting the recruitment of TAMs, via different signaling pathways, has proven beneficial at the preclinical level and is now being tested in several cancer clinical trials for cancer [93]. Despite evidence for this strategy in BCa, no clinical study is currently targeting this tumor type.

### 4.2. Macrophage Depletion for Bladder Cancer Treatment

#### 4.2.1. CSF1-CSF1 R Axis

Another well-investigated strategy toward TAMs is their depletion. CSF1 is a growth factor essential for the proliferation, differentiation, survival and recruitment of bone marrow-originated macrophages. This makes the CSF1-CSF1 R pathway an interesting target to deplete macrophages. CSF1 R inhibition, with small molecules or neutralizing antibodies, results in macrophage reduction and increased mouse survival in models of fibrosarcome, melanoma and mammary, colon and pancreatic tumors [158,159,160,161]. However, the relevance of blocking the CSF1-CSF1 R pathway in BCa is not yet clear as only a few studies have analyzed this axis and without convincing results. Even if CSF1 was secreted by bladder tumor cells [162,163], only 34% of patients have a high expression of CSF1 in tumor tissues. Moreover, CSF1 expression was neither associated with clinicopathological features nor correlated with RFS or CSS [164]. However, CSF1 serum and urine levels of MIBC patients were higher than those of NMIBC patients and controls [164].

#### 4.2.2. Trabectedin

Another method found to specifically deplete macrophages is Trabectedin. Trabectedin, a DNA binder of marine origin, was first used as a chemotherapy for leukemia. However, preclinical and clinical analyses have demonstrated its efficacy in depleting monocytes and MDSCs in blood, macrophages in spleen and TAMs, without affecting T cells or neutrophils [165,166]. TAM reduction following Trabectedin results in decreased angiogenesis and metastasis development [167,168,169] but also induces a better immunotherapy efficacy in a mouse model of chronic lymphoid leukemia [166]. Germano and coworkers demonstrated that Trabectedin induced macrophage apoptosis through a pathway involving the TNF-related apoptosis-inducing ligand (TRAIL) receptors 1 and 2, which are expressed on monocytes and macrophages but not on T cells and neutrophils [165]. This recent strategy is appealing in targeting TAMs; however, no study has reported trying this molecule in BCa.

#### 4.2.3. Bisphosphonates

Bisphosphonates are inorganic compounds that include clodronate. Even if bisphosphonates were first used as anticancer agents for hematologic and solid tumors, it was shown that they decrease proliferation, migration and invasion of macrophages, resulting in their apoptosis [170]. A classical approach to use bisphosphonates to target macrophages is to encapsulate clodronate in liposomes, which will be preferentially taken up by macrophages due to their phagocytic properties. This strategy was demonstrated as efficient in depleting monocytes, resulting in TAM reduction, accompanied by the decrease of tumor size, angiogenesis and metastasis development in different tumor models [64,171,172]. TAM depletion by clodronate liposomes improves the efficacy of chemotherapy [173] and anti-angiogenic therapy [174]. In the MBT-2 model of BCa, TAM depletion reduced lymphangiogenesis, and in consequence, some lymphatic metastases. However, we demonstrated that the depletion of TAMs in a MIBC model decreases the efficacy of the anti-CD40/anti-PD-1 combination therapy [39], suggesting that the therapy-induced M1-like TAMs are essential for the response to the therapy itself. Our results were in accordance with the study of Klug et al., which indicated that M1-like TAMs were required for the beneficial effect of low dose irradiation and that clodronate-induced TAM depletion inhibited the positive response of the treatment [175]. These strategies are evaluated in different clinical trials [93] but not specifically for BCa. Moreover, even if TAM depletion is an attractive strategy, it will not conserve M1-like TAMs, which are important for the anti-tumor response. For this reason, TAM-targeting strategies now try to reprogram/deplete M2-like TAMs and enhance the M1-like TAMs.

### 4.3. Reprogramming of Tumor-Associated Macrophages in Bladder Cancer

#### 4.3.1. Chemokines

Inhibitors or blocking antibodies for certain chemokines described above were shown to efficiently reprogram TAMs, instead of depleting them or inhibiting their recruitment. BLZ945, a small molecule against CSF1 R, reprogramed TAMs in brain tumors, resulting in increased survival [176]. The combination of TAM reprograming following BLZ945 treatment and radiotherapy significantly prolonged survival in glioma-bearing mice [177]. A reduction of M2-like TAMs was also observed with an anti-CSF1 R blocking antibody in a model of pancreatic ductal adenocarcinoma, additionally improving the efficacy of ICIs [178]. In a model of prostate cancer, an anti-CXCR2 blocking antibody induced the reeducation of TAMs, resulting in a decrease in tumor volume [179]. 

#### 4.3.2. Toll-like Receptor (TLR) Agonists

TLRs are a family of PRRs that are fundamental for the activation of innate immune cells [180]. TLR engagement results in nuclear factor-κB (NF-κB) translocation into the nucleus, which will induce the expression of inflammatory genes [181]. Several in vitro experiments have demonstrated that TLR engagement with TLR agonists reprograms M2-like macrophages toward the M1-like phenotype [182,183,184,185,186,187]. Among all TLR agonists, the first FDA-approved was BCG. It is recognized by TLR2/4/9 on urothelial and immune cells and induces the secretion of pro-inflammatory cytokines [94]. As discussed above, BCG treatment increases macrophage infiltration and their cytotoxic activity, supporting the use of TLR agonists in TAM reprogramming. Imiquimod, a TLR7 agonist, was also efficient in inducing an intense local inflammation and a decrease in tumor growth in models of BCa [188,189,190]. A clinical study on NMIBC patients indicated that a TLR7 agonist, in combination with BCG, induced a significant increase of cytokines in the urine of patients and increased clinical responses [191]. The same report was noted for polyinosinic:polycytidylic acid (poly(I:C)), a TLR3 agonist, also in combination with BCG in models of BCa [192,193]. Moreover, poly(I:C) treatment was efficient to enhance the anti-PD-1 response in a NMIBC model [193]. These results indicate that TLR agonists could be further investigated for the treatment of BCa to complement or replace BCG in NMIBC, or to improve ICIs in advanced BCa. Different agonists are currently being tested in phase I and II clinical trials for BCa (Table 3).

#### 4.3.3. Histone Deacetylase (HDAC) Inhibitors

HDACs are enzymes implicated in the epigenetic regulation of gene expression and are responsible for the removal the acetyl groups on histones. HDAC inhibitors are the first class of epigenetic drugs to be FDA-approved for cancer therapy and can be classified according to their HDAC specificity [194]. They can inhibit the deacetylation of histones or nonhistone proteins and have direct effects on tumor cells, as well as immune cells. TMP195, a selective competitive class IIa HDAC inhibitor, has shown to alter monocyte/macrophage gene expression without affecting lymphocytes [195]. In vitro and in vivo studies have reported that TMP195 reprograms M2-like macrophages toward M1-like macrophages, resulting in the reduction of tumor growth [124,195]. The authors demonstrated that M1-like macrophages, IFNγ and CD8^+^ T cells were required for the anti-tumor effect of the TMP195 treatment [124]. This phenomena of TAM reprogramming was also demonstrated with other HDAC inhibitors [196,197]. However, it has been proposed that the reduction in M2-like TAMs after HDAC inhibition could be an indirect effect. HDAC inhibition could downregulate CCR2 expression, resulting in a decrease of MDSCs and hence skew the M-MDSC-to-macrophage differentiation [198]. In different tumor models, TAM reprograming via HDAC inhibitors enhances the efficacy of anti-PD1 and anti-PD-L1 blockade antibodies [124,197,199]. In BCa, several HDAC inhibitors have been tested as anti-tumor agents resulting in bladder tumor cell cytotoxicity [200,201,202,203]. Only one preclinical study has demonstrated that Vorinostat, a pan-HDAC inhibitor, can modify the tumor microenvironment of the MB49 NMIBC model [204]. In this study, the authors declared that Vorinostat treatment enhances the anti-PD-1 response and that the combination therapy was effective at inducing CD8^+^ T cell recruitment. Even if TAM numbers were unaffected after HDAC inhibition, their phenotype and their role in the anti-tumor response of the treatment was not investigated [204]. Two HDAC inhibitors, Mocetinostat and Vorinostat, have been administered to patients with metastatic BCa (NCT02236195 and NCT00363883) but both treatments were associated with limited efficacy and significant toxicity [205,206]. Even if the first clinical trials in BCa present limited results, HDAC inhibitors seem promising to reprogram TAMs and others are currently being tested in BCa (Table 3).

#### 4.3.4. Phosphoinositide 3-Kinase (PI3 K) Inhibitors

PI3 K is involved in almost all types of cell signaling and is divided in several subclasses, from which the class IB isoform PI3 Kγ is mainly expressed by hematopoietic cells [207]. It was shown that the inhibition of PI3 Kγ results in impaired recruitment of myeloid cells, mainly macrophages and neutrophils, to the tumor site [208]. More recently, Varner’s lab has demonstrated that the inhibition of PI3 Kγ could reprogram M2-like myeloid cells toward an M1-like phenotype resulting in the recruitment of cytotoxic T cells and the reduction of tumor growth [209,210,211]. This strategy also enhanced the efficacy of chemotherapy and ICIs in different tumor models [209,210]. In BCa, no study has investigated the inhibition of PI3 Kγ in TAMs. It seems that the PI3 K signaling pathway is essential in bladder tumor cells, regulating their proliferation, migration, invasiveness and metastasis [212]. For that, several PI3 K inhibitors have been tested in clinical trials (Table 3).

For the moment, the molecules that could reprogram TAMs are being tested because of their direct effect on tumor cells. It seems that reprograming TAMs toward an anti-tumor phenotype is the most interesting method but it appears to be more complicated than what has been observed in vitro. Due to this, another way to target TAMs is to activate their anti-tumor functions via the activation of stimulatory molecules or the blocking of inhibitory molecules.

### 4.4. Activation of Tumor-Associated Macrophages in Bladder Cancer

#### 4.4.1. CD40-CD40 L Pathway

CD40 is a receptor of the TNF receptor family that is widely expressed on antigen-presenting cells. The CD40-CD40 L interaction is important in cross-priming T cells and consequently for the amplification and regulation of the inflammatory response [213]. Based on that, Beatty et al. have demonstrated that an agonistic CD40 antibody activated TAMs in a model of pancreatic cancer, induced tumor regression and enhanced the efficacy of chemotherapy [214]. These results were confirmed in patients with pancreatic ductal adenocarcinoma. This demonstrated for the first time the efficacy of anti-CD40-activated TAMs [214]. Several preclinical studies have then followed to demonstrate the efficacy of anti-CD40 therapy in stimulating TAMs and resulting in tumor regression [215,216,217] and improvement of chemotherapy [218,219], anti-angiogenic therapy [217] and ICI [220,221] efficacies in different tumor models. Anti-CD40 immunotherapy has also been tested in several BCa preclinical studies. However, it seems that its efficacy was mediated by the activation of dendritic cells (DCs) instead of TAMs. In bladder tumors, CD40 is mainly expressed by DCs and MHCII^+^ TAMs [222]. Its expression decreases with tumor progression [39], confirming the need to revitalize antigen-presenting cells via this pathway. In the MB49 model, an anti-CD40 agonist antibody activates DCs and then reverses CD8^+^ T cell exhaustion signatures, resulting in the reduction of tumor burden and increase of survival [222,223]. Garris and coworkers also demonstrated that the intravesical delivery of anti-CD40 antibody induces local anti-tumor activity in a BCG-unresponsive BCa model [222]. However, in an anti-PD-1-resistant model of MIBC, we demonstrated that anti-CD40 therapy induced an anti-PD-1 response by activating DCs in tumor-draining lymph nodes but not in the tumor [39]. This resulted in the activation of CD8^+^ T cells in tumor-draining lymph nodes, egress of the T cells from the lymph nodes and infiltration into the bladder tumor. The effect of anti-CD40 antibodies to induce a systemic inflammation was confirmed by Sandin et al., who revealed that the CD40-specific antibody accumulated in the bladder tumor-draining lymph nodes and the spleen [224]; organs where antigen-presenting cells are abundant. Even if TAMs did not seem to be the direct targets of anti-CD40 antibodies in BCa, we demonstrated that the critical CD8^+^ T cell activation generated a switch from M2-like to M1-like TAMs and this was important for the anti-CD40^+^ anti-PD-1 combination therapy [39]. These studies indicate that TAM activation via CD40 agonists is an appealing strategy for BCa treatment and several are being tested in clinical trials (Table 3).

#### 4.4.2. CD47-Signal Regulatory Protein-α (SIRPα) Axis 

CD47 is a transmembrane protein found ubiquitously expressed on normal cells as a “self” marker, but it is also overexpressed by tumor cells. CD47 can bind to SIRPα, which is mainly expressed on phagocytic myeloid cells. CD47-SIRPα engagement results in a “do not eat me” signal that provides an immune escape pathway for tumor cells [225]. Blocking the CD47-SIRPα pathway with anti-CD47 blocking antibodies increases phagocytosis of tumor cells by macrophages in vitro [226,227,228,229]. It seems that anti-CD47 blockade increases the phagocytosis of tumor cells [229] and the secretion of cytokines and chemokines, which promote macrophage recruitment [227]. Anti-CD47 treatment elevates the number of TAMs in an ovarian cancer model, yet still resulting in a reduction of tumor cell numbers and an increase in survival [228]. The beneficial effect of anti-CD47 immunotherapy was confirmed in other tumor models [226,227,230]. In the case of BCa, it was shown that CD47 was expressed by at least 80% of tumor cells [230,231]. Incubation with an anti-CD47 antibody induced phagocytosis of bladder tumor cells by macrophages, inhibiting primary tumor growth and preventing tumor metastasis in vivo [230,231]. Recently, Kiss et al. combined an anti-CD47 antibody with an infrared dye to perform near-infrared photoimmunotherapy (NIR-PIT) in a model of BCa. They demonstrated that the NIR-PIT had the advantage of inducing direct tumor cell death and enhancing macrophage phagocytic abilities, which resulted in a higher TAM density and slower tumor growth compared to monotherapy [232]. Blocking this innate immune checkpoint is a strategy that is now being tested in clinical trials as a monotherapy for hematologic tumors and as combination therapy for solid tumors [93] and is starting to be investigated in the context of BCa (Table 3).

## 5. Conclusions

In this review, we provided evidence of the presence and importance of macrophages in BCa. Pro-tumor macrophages influence the stage of the disease, are associated with poor patient outcome, and play deleterious roles in ongoing treatments. Several macrophage-targeting strategies are promising for future BCa treatment, but only a few investigations have explored this possibility for BCa immunotherapy. A better characterization of TAMs in BCa remains necessary to establish the best targets, both in preclinical models and in patients. We believe that in a near future, macrophage-targeting therapies can be used in adjuvant or neo-adjuvant settings for current treatments, i.e., surgery and chemotherapy, but also in combination with immunotherapy to increase the efficacy of PD-1/PD-L1 blocking antibodies to benefit the patients.

## Figures and Tables

**Figure 1 cancers-13-04712-f001:**
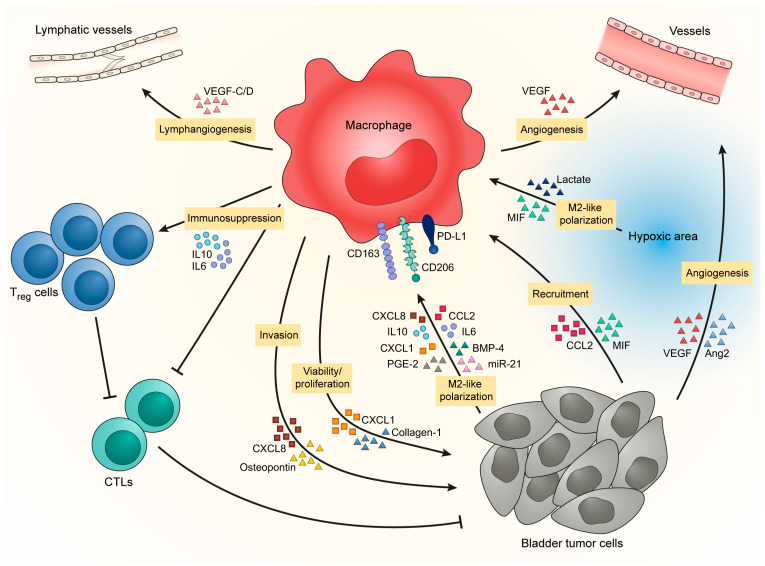
BCa favors M2-like polarization of TAM to promote tumor progression.

**Figure 2 cancers-13-04712-f002:**
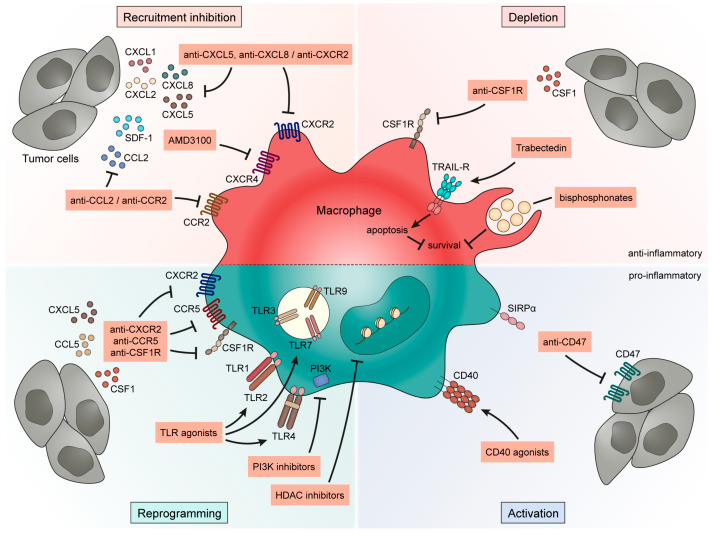
Therapeutic strategies to target TAMs in BCa.

**Table 1 cancers-13-04712-t001:** Macrophages and bladder cancer clinical outcomes.

Cell Types and Markers	Bladder Cancer Cohorts	Types of Sample	Findings	References
CD68^+^ (TAM)	40 pTa-pT1 23 ≥ pT2	FFPE tissue	High TAM count was associated with poor 5-year survival	Hanada et al. [29]
CD68^+^ (TAM)	81 pTa-pT1 11 pT2	FFPE tissue	High number of TAMs was significantly associated with risk of progression	Bostrom et al. [31]
CD68^+^ (TAM)	112 pTa 89 pT1 93 MIBC	FFPE tissue	High CD68/CD3 ratio was associated with poor OS	Sjödahl et al. [45]
CD68^+^ (TAM)	10 low grades 34 high grades	FFPE tissue	High TAM count was associated with poor DFS but not OS	Chai et al. [50]
CD163^+^ (TAM)	115 MIBC	FFPE tissue	High TAM count was associated with poor PFS and OS	Xu et al. [35]
CD163^+^ (TAM)	94 high grade pT1	FFPE tissue	TAMs were associated with tumor recurrence and progression	Yang et al. [43]
CD204^+^round cells (TAM)	155 NMIBC	FFPE tissue	High number of TAMs correlated with a high risk of recurrence	Miyake et al. [42]
TAM	MIBC (no metastatic disease)	TCGA database	A signature with low T cells, low NK cells, high Treg and high TAMs is associated with poor DFS and OS	Fu et al. [24]
TAM	406 MIBC	TCGA database	Patients in the high-risk group had a signature with low CD8^+^ T cells, CD4^+^ T cells and high abundance of M0 macrophages	Li et al. [44]
CD204^+^CD68^+^ (CD204^+^ macrophages)	212 pTa-pT1 90 pT2-pT4	FFPE tissue	High number of CD204^+^ macrophages in tumor stroma was associated with poor OS	Wang et al. [25]
DC-SIGN^+^CD68^+^ (DC-SIGN^+^TAM)	257 MIBC	FFPE tissue	DC-SIGN^+^ TAMs may contribute to progression and poor prognosis	Hu et al. [46]
M2-like TAM	429 MIBC	TCGA database	High M2-like TAM signature was associated with poor OS and DFS	Xue et al. [38]
M2-like TAM	402 MIBC	TCGA database	M2-like TAM signature was associated with significantly worse 5-year OS and DFS outcomes	Jiang et al. [47]
CD169^+^CD68^+^ (CD169^+^macrophages)	44 MIBC	FFPE tissue	CD169^+^ macrophages in tumor-draining lymph nodes were positively correlated with a favorable prognosis	Asano et al. [48]
CD33^+^ (MDSC)	70 NMIBC 27 MIBC	FFPE tissue	The number of tumor-infiltrating CD33^+^ MDSCs was significantly inversely correlated with patient OS	Zhang et al. [27]
CD11b^+^CD33^low^HLA-DR^−^ (MDSC)	71 pTa-pT1 42 pT2-pT4	PBMC	High numbers of circulating CD11b^+^ CD33^low^HLA-DR^−^ cells were correlated with poor OS	Yang et al. [32]

CSS: cancer-specific survival; DFS: disease-free survival; FFPE: formalin-fixed paraffin-embedded; MDSC: myeloid-derived suppressor cell; MIBC: muscle-invasive bladder cancer; NK cells: natural killer cells; NMIBC: nonmuscle-invasive bladder cancer; OS: overall survival; PBMC: peripheral blood mononuclear cell; PFS: progression-free survival; TAM: tumor-associated macrophage; TCGA: The Cancer Genome Atlas; Treg: regulatory T cells.

**Table 2 cancers-13-04712-t002:** Macrophages and responses to bladder cancer treatments.

Cell Types and Markers.	Bladder Cancer Cohorts	Treatments	Findings	References
CD68^+^ (TAM)	40 pTa-pT1 23 ≥ pT2	TURBT or RC	Patients with a high TAM count showed higher rates of cystectomy than those with a low TAM count	Hanada et al. [29]
CD68^+^ (TAM) and MAC387^+^ CD68^+^ (MAC387^+^ TAM)	81 pTa-pT1 11 pT2	TURBT or RC	TURBT tumors have lower TAM counts than RC tumors High TAM counts were associated with poorer survival in TURBT patients MAC387+ CD68+ cells were associated with poorer survival in RC patients	Bostrom et al. [31]
CD68^+^ (macrophage) and CD204^+^ CD68^+^ (CD204^+^macrophages)	212 pTa-pT1 90 pT2-pT4	TURBT or RC	Total macrophages and CD204^+^ macrophages in the stroma were associated with poor OS after surgery	Wang et al. [25]
CD68^+^ (TAM) and HIF-2α^+^ CD68^+^ (HIF-2α^+^ TAM)	22 pT1 20 pT2 23 pT3 4 pT4	RC	No significant association between TAM indexes and the prognosis in patients undergoing RC HIF-2α^+^ TAMs were associated with a poor prognosis after RC	Koga et al. [54]
CD33^+^ HLA-DR^−^ (MDSC)	65 pTa-pT1 44 ≥ pT2	RC	The percentage of total MDSC in PBMC before RC was significantly lower in patients who experienced pathological complete response	Fallah et al. [77]
CD68^+^ (TAM)	3 pTa 9 pT1 18 pTis	BCG	Higher CD68^+^ cells in tumor after BCG are correlated with better RFS	Kitamura et al. [78]
CD68^+^ (TAM)	53 NMIBC	BCG	High TAM is associated with poor RFS in high-risk NMIBC after BCG	Ayari et al. [30]
CD68^+^ (TAM)	41 CIS	BCG	Low TAM count is associated with good RFS after BCG	Takayama et al. [79]
CD68^+^ (TAM)	12 pTa 15 pT1	BCG	High TAM count is associated with shorter RFS after BCG treatment	Aliji et al. [80]
CD68^+^ (TAM)	304 NMIBC	BCG	Pre-BCG treatment TAMs are associated with worse RFS in patients with NMIBC	Kardoust et al. [81]
CD204^+^ (TAM)	68 pTa 73 pT1 13 pTis	BCG	High counts of TAM showed association with short PFS after BCG	Miyake et al. [34]
CD163^+^ CD68^+^ (CD163^+^ macrophages)	40 pTa 59 pT1	BCG	High density of CD163^+^ macrophage counts in the stroma but not in the tumor was related with BCG failures	Lima et al. [26]
iNOS^+^ CD68^+^ (iNOS^+^ TAM) and CD163^+^ CD68^+^ (CD163^+^ TAM)	40 NMIBC	BCG	High iNOS^+^ TAM counts were associated with better DFS after BCG instillation High CD163^+^ TAM counts were associated with poor DFS after BCG instillation	Suriano et al. [82]
CD68^+^ (TAM) and CD163^+^ CD68^+^ (CD163^+^ TAM)	9 pTa 21 pT1 10 pTis	BCG	The median number of total CD68^+^ TAMs and CD163^+^ TAMs were significantly increased in patients with BCG failure compared to BCG responders High numbers of CD68^+^ TAMs, high numbers of CD163^+^ TAMs and a high CD163/CD68 ratio were associated with a greater risk of recurrence after BCG	Pichler et al. [83]
Lin^−^CD14^+^ CD33^+^ HLA-DR^−^ (M-MDSC)	4 pTa 20 pT1 3 pTis 1 pT2	BCG	Low T cell/M-MDSC ratio after BCG treatment correlates with poor RFS & PFS	Chevalier et al. [41]
CD68^+^ (TAM)	49 pT2 69 ≥ pT3	platinum-based chemotherapy	An immunotype containing low T cells, low NK cells, high Treg and high TAM is associated with increase OS and DFS after chemotherapy in pT3-T4 patients	Fu et al. [24]
DC-SIGN^+^ CD68^+^ (DC-SIGN^+^ TAM)	137 pT2	cisplatin-based chemotherapy	High DC-SIGN^+^ TAM infiltration was strongly associated with unresponsiveness to adjuvant chemotherapy in MIBC	Hu et al. [46]
Galectin-9^+^ CD68^+^ (Gal9^+^ TAM)	141 ≥ vpT2	platinum-based chemotherapy	Survival benefits after postoperative adjuvant chemotherapy among patients with high Gal9^+^ TAM, whereas patients with low Gal9^+^ TAM showed no benefit to chemotherapy	Qi et al. [73]
CD68^+^ (TAM) and CD163^+^CD68^+^ (CD163^+^TAM)	44 pT2 85 pT3 39 pT4	Adjuvant chemotherapy	Chemotherapy was associated with a longer OS and DSS and showed a trend with a longer RFS in pT3 and pT4 patients with low CD68 expression Chemotherapy was associated with a trend toward longer OS compared with no chemotherapy in pT3 and pT4 patients with low CD163 expression	Taubert et al. [84]
CD33^+^ HLA-DR^−^ (MDSC)	49 < pT2 36 ≥ pT2	Neoadjuvant chemotherapy	Circulating MDSCs were negatively associated with pathologic complete response in patients treated with neoadjuvant therapy	Ornstein et al. [85]
PD-L1^+^ tumor-infiltrating immune cells	Metastatic urothelial carcinoma (IMvigor210)	Atezolizumab	PD-L1 expression on immune cells was significantly associated with response to Atezolizumab	Mariathasan et al. [86]
PD-L1^+^ tumor-infiltrating immune cells	Metastatic urothelial bladder cancer (IMvigor211)	Atezolizumab	Tumors expressing PD-L1^+^ tumor-infiltrating immune cells had particularly high response rates	Powles et al. [87]
PD-L1^+^ tumor-infiltrating immune cells	Metastatic urothelial carcinoma	Atezolizumab	Higher levels of PD-L1 immunohistochemistry expression on immune cells were associated with a higher response rate to Atezolizumab and longer OS	Rosenberg et al. [88]
PD-L1^+^ tumor-infiltrating immune cells	Metastatic urothelial bladder cancer (IMvigor211)	Atezolizumab	Overexpression of PD-L1 resulted in a more favorable outcome with both chemotherapy and Atezolizumab	Powles et al. [89]
PD-L1^+^ tumor-infiltrating immune cells	Advanced urothelial cancer (KEYNOTE-045)	Pembrolizumab	The benefit of Pembrolizumab appeared to be independent of PD-L1 expression on infiltrating immune cells	Bellmunt et al. [90]
PD-L1^+^ cells	Unresectable locally advanced or metastatic urothelial carcinoma (CheckMate 032)	Nivolumab + Ipilimumab	Responses were observed regardless of PD-L1 expression levels	Sharma et al. [91]
M1-like TAM	Metastatic urothelial carcinoma (IMvigor210)	Atezolizumab	M1 frequency is a robust biomarker for predicting the prognosis and response to immune checkpoint blockades	Zeng et al. [92]
Pro-tumorigenic inflammation signature	Metastatic urothelial carcinoma (IMvigor210)	Atezolizumab	Pro-tumorigenic inflammation in individual tumor microenvironments is associated with PD-1 and PD-L1 resistance	Wang et al. [93]
	Metastatic urothelial carcinoma (CheckMate 275)	Nivolumab		

BCG: Bacillus Calmette-Guérin; DFS: disease-free survival; FFPE: formalin-fixed paraffin-embedded; MDSC: myeloid-derived suppressor cell; M-MDSC: monocytic-MDSC; MIBC: muscle-invasive bladder cancer; NK cells: natural killer cells; NMIBC: nonmuscle-invasive bladder cancer; OS: overall survival; PBMC: peripheral blood mononuclear cell; PFS: progression-free survival; RC: radical cystectomy; RFS: recurrence-free survival; TAM: tumor-associated macrophage; T cells: lymphocytes; TCGA: The Cancer Genome Atlas; Treg: regulatory T cells; TURBT: trans-urethral resection of bladder tumor.

**Table 3 cancers-13-04712-t003:** Clinical trials including bladder cancer with agents targeting macrophages.

Targeted Pathways	Agent Names	Combinations	Tumor Types	Clinical Phases	Trial Numbers
TLR	BDB001	Atezolizumab + Radiotherapy	Advanced solid tumors *	II	NCT03915678
	Imiquimod		Carcinoma in situ bladder cancer	II	NCT01731652
	Imiquimod	TRK-950	Advanced solid tumors *	I	NCT03872947
	Poly(I:C)	PGV001 + Atezolizumab	Urothelial/bladder cancer	I	NCT03359239
	Poly(I:C)	Durvalumab +/− Tremelimumab	Advanced solid tumors *	I/II	NCT02643303
HDAC	Abexinostat	Pembrolizumab	Advanced solid tumor *	I	NCT03590054
	Belinostat		Bladder cancer	I/II	NCT00421889
	Chidamide	Tislelizumab	Bladder cancer stage IV	II	NCT04562311
	Domatinostat	Nivolumab +/− Ipilimumab	Urothelial carcinoma	I	NCT04871594
	Entinostat	Pembrolizumab	MIBC	II	NCT03978624
	FR901228		Advanced urothelial carcinoma	II	NCT00087295
	Mocetinostat		Urothelial carcinoma	II	NCT02236195
	Romidepsin		Solid tumors *	I	NCT01638533
	Vorinostat		Locally recurrent or metastatic urothelial carcinoma	II	NCT00363883
	Vorinostat	Docetaxel	Advanced and relapsed solid tumors *	I	NCT00565227
	Vorinostat	Pembrolizumab	Advanced urothelial cell carcinoma	I	NCT02619253
PI3K	Buparlisib		Metastatic urothelial carcinoma	II	NCT01551030
	Copanlisib Copanlisib Hydrochloride GSK2636771 Taselisib		Advanced solid tumors *	II	NCT02465060
	Eganelisib	Nivolumab	Advanced urothelial carcinoma	II	NCT03980041
CD40	APX005M		Urothelial carcinoma	I	NCT02482168
	CDX-1140	+/− CDX-301 +/− Pembrolizumab +/− Chemotherapy	Advanced solid tumors *	I	NCT03329950
CD47	Hu5F9-G4	Atezolizumab	Cisplatin-ineligible MIBC and locally advanced or metastatic urothelial carcinoma	I/II	NCT03869190

*: in tumor types, the study specified that bladder cancer or urothelial carcinoma are included. HDAC: histone deacetylase; PI3 K: phosphoinositide 3-kinase; TLR: Toll-like receptor.

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
