# Peer review of "Tumor-Associated Macrophages in Bladder Cancer: Biological Role, Impact on Therapeutic Response and Perspectives for Immunotherapy"

_cancers, 2021, doi:10.3390/cancers13184712_

Round 1
Reviewer 1 Report
The review paper describes the contribution of macrophages to bladder cancer progression and proposes future immunotherapeutic strategies by reducing macrophages’ functions. Although the article provides a useful review of tumor-associated macrophages in bladder cancer, I would like to ask for several modifications and reconstructions in the manuscript before publication.
- The abstract does not match well the content of the text. The authors attempted to cover the whole area of macrophages and the bladder, but it resulted in a blurred manuscript. Writing sections 1 and 2 short and concise, even omit, or starting with section 3 will improve the manuscript.
- The description in section 1.3. should be provided with that of cell surface markers. If possible, the authors insert a Table showing each TAM subset with its functions and corresponding markers. The whole picture about various TAM subsets (for example, Galectin-9+, MAC387+, HIF-2a+, DC-SIGN+, CD169+….) is not well organized, therefore the manuscript is difficult to read.
- Section 3.4 (TAM recruitment) can be combined with section 5.1 or described in another section.
- Section 4 should be written concisely with a Table, as section 3.2.
- Section 5, Table 2: It is sufficient to show the studies relevant for bladder/urothelial carcinoma.
- No figure title on page 8, whereas the wrong title is with Figure 2 on page 13.
Author Response
Cancers-1366630, point-by-point reply
Manuscript by M. Leblond et al. previously entitled: ‘Targeting tumor-associated macrophages: a tempting perspective for bladder cancer immunotherapy’
Now entitled “Tumor-associated macrophages in bladder cancer: biological role, impact on therapeutic response and perspectives for immunotherapy”
Dear Editor and Reviewers,
We thank you for the insightful comments on our review. We believe they improved our review and made it clearer and easier to read. Please find our point-by-point reply. As major changes were made, the manuscript version is without track changes to facilitate the reading. We hope that our new version complies with the standards for publication in Cancers and thank you for your kind consideration.
Reviewer #1.
The review paper describes the contribution of macrophages to bladder cancer progression and proposes future immunotherapeutic strategies by reducing macrophages’ functions. Although the article provides a useful review of tumor-associated macrophages in bladder cancer, I would like to ask for several modifications and reconstructions in the manuscript before publication.
- The abstract does not match well the content of the text. The authors attempted to cover the whole area of macrophages and the bladder, but it resulted in a blurred manuscript. Writing sections 1 and 2 short and concise, even omit, or starting with section 3 will improve the manuscript.
We thank the reviewer for this comment, and we agree with this point. We removed a large part of sections 1 and 2 and wrote a shorter part 1 (as introduction) focused on 1.1-TAMs and 1.2-bladder cancer that will be followed directly by our previous section 3 (now section 2). The abstract was adapted to these changes.
- The description in section 1.3. should be provided with that of cell surface markers. If possible, the authors insert a Table showing each TAM subset with its functions and corresponding markers. The whole picture about various TAM subsets (for example, Galectin-9+, MAC387+, HIF-2a+, DC-SIGN+, CD169+….) is not well organized, therefore the manuscript is difficult to read.
We understand the reviewer’s concern and we apologize for not making this point clear. Several reviews already recapitulate M1 and M2 markers used for macrophages and TAM (references are included in our manuscript: Jayasingam et al., Front. Oncol., 2020; Rath et al., Front. Immunol., 2014). Most of the studies we report here used only 1 or 2 markers, which do not allow to determine whether these various TAM populations are overlapping or not. However, to make this point clearer for readers, we describe the various markers in part 1.1 and we have also updated Table 1 to include relevant markers used by the presented studies. We believe that this kind of representation is suitable for the message of this review. We do not believe a detailed table of the markers and functions is appropriate for this review.
- Section 3.4 (TAM recruitment) can be combined with section 5.1 or described in another section.
We thank the reviewer for this suggestion. We described TAM recruitment in a specific section (2.2 in the revised version).
- Section 4 should be written concisely with a Table, as section 3.2.
We thank the reviewer for this comment. We now included a new table (Table 2 in our revised version) that recapitulates the results from section 4 (now section 3) and we made this part more concise.
- Section 5, Table 2: It is sufficient to show the studies relevant for bladder/urothelial carcinoma.
We thank the reviewer for this suggestion. We removed the clinical trials that did not specifically include bladder cancer or urothelial carcinoma.
- No figure title on page 8, whereas the wrong title is with Figure 2 on page 13.
We apologize for this. The editor informed us that there was a problem with the formatting of the manuscript before it was sent to reviewers. We fixed the problems in our revised manuscript.
Reviewer 2 Report
Authors of the present narrative review aimed at summarizing current evidence about the prognostic and potential therapeutic role of tumor-associated macrophages in bladder cancer. The issue has relevant pre-clinical and clinical implications and deserves consideration for publication. However, the manuscript should undergo major revisions to improve the overall quality.
- Title: in the current form, the title of the review focuses only on the therapeutical implications of tumor-associated macrophages and does not take into account all the issues addressed in the reviews (for example prognostic implications).
- Paragraph 1 (Generalities on macrophages) and paragraph 2 (macrophages and the healthy bladder) are too dispersive. I suggest synthesizing the main concepts expressed in these paragraphs together with data provided in subparagraph 3.1 (Bladder Cancer) in a single "Introduction" paragraph and then move to the core of the review.
- Paragraph 4 (Macrophages influence bladder cancer treatments): I suggest to split the paragraph according to the treatments for non-muscle invasive and muscle invasive bladder cancer.
- Tables 1 and 2 should be improved graphically (they seem cut in the lower part).
- The description of the Figure reported on page 8 is missing.
- Figure reported on page 13 ( the second figure in the manuscript) is described as Figure 1.
- I suggest to add a paragraph on "future perspectives"
- I suggest to use the acronym "BCa" for bladder cancer
Author Response
Cancers-1366630, point-by-point reply
Manuscript by M. Leblond et al. previously entitled: ‘Targeting tumor-associated macrophages: a tempting perspective for bladder cancer immunotherapy’
Now entitled “Tumor-associated macrophages in bladder cancer: biological role, impact on therapeutic response and perspectives for immunotherapy”
Dear Editor and Reviewers,
We thank you for the insightful comments on our review. We believe they improved our review and made it clearer and easier to read. Please find our point-by-point reply. As major changes were made, the manuscript version is without track changes to facilitate the reading. We hope that our new version complies with the standards for publication in Cancers and thank you for your kind consideration.
Reviewer #2.
Authors of the present narrative review aimed at summarizing current evidence about the prognostic and potential therapeutic role of tumor-associated macrophages in bladder cancer. The issue has relevant pre-clinical and clinical implications and deserves consideration for publication. However, the manuscript should undergo major revisions to improve the overall quality.
- Title: in the current form, the title of the review focuses only on the therapeutical implications of tumor-associated macrophages and does not take into account all the issues addressed in the reviews (for example prognostic implications).
We understand the reviewer’s concern. We changed the title to “Tumor-associated macrophages in bladder cancer: biological role, impact on therapeutic response and perspectives for immunotherapy“ to address all the points raised in this review.
- Paragraph 1 (Generalities on macrophages) and paragraph 2 (macrophages and the healthy bladder) are too dispersive. I suggest synthesizing the main concepts expressed in these paragraphs together with data provided in subparagraph 3.1 (Bladder Cancer) in a single "Introduction" paragraph and then move to the core of the review.
We thank the reviewer for this comment, and we agree with this point. We removed a large part of sections 1 and 2 and wrote a shorter part 1 (as introduction) focused on 1.1-TAMs and 1.2-bladder cancer that will be followed directly by our previous section 3 (now section 2).
- Paragraph 4 (Macrophages influence bladder cancer treatments): I suggest to split the paragraph according to the treatments for non-muscle invasive and muscle invasive bladder cancer.
We thank the reviewer for this suggestion. This is now done in the revised manuscript (section 3 in our revised version).
- Tables 1 and 2 should be improved graphically (they seem cut in the lower part).
We apologize for this. The editor informed us that there was a problem with the formatting of the manuscript before it was sent to reviewers. We fixed the problems in our revised manuscript. This was also the case for points 6 and 7.
- The description of the Figure reported on page 8 is missing.
- Figure reported on page 13 ( the second figure in the manuscript) is described as Figure 1.
- I suggest to add a paragraph on "future perspectives"
We thank the reviewer for this suggestion. We modified the conclusion paragraph in our revised version to include some future perspectives. We believe that a much better characterization of TAMs in BCa is necessary before really precisely stating the future perspectives. But we are convinced that targeting this cell type is promising for BCa treatments.
- I suggest to use the acronym "BCa" for bladder cancer
We thank the reviewer for this suggestion. We use the acronym “BCa” for bladder cancer in our revised version.
Reviewer 3 Report
The review by Leblond et al., „Targeting tumor-associated macrophages: a tempting perspective for bladder cancer immunotherapy” is very timely and provides a comprehensive overview about macrophages in bladder cancer, link tumor-associated macrophages to clinical outcomes as well as how macrophages are affected by different anti-cancer treatment regimens. The figures and tables are exceptionally well designed.
Minor comments:
- Line 303: scRNA-seq is only briefly mentioned. It would very informative to add a whole paragraph to this chapter and provide more insight into the diversity/complexity of TAMs in bladder cancer as revealed by scRNA-seq. Several other scRNA-seq papers could be included, e.g. https://www.biorxiv.org/content/10.1101/2020.07.09.195685v1.full. How do bladder-resident macrophages compare to TAMs in bladder cancer? Are TAMs in bladder cancer different to TAMs in other cancers? (e.g. does one find SPP1, C1QC TAMs also in bladder cancer?).
It seems that chapter 3.3 is missing or chapter 3.4 needs to be re-labeled to 3.3.
Author Response
Cancers-1366630, point-by-point reply
Manuscript by M. Leblond et al. previously entitled: ‘Targeting tumor-associated macrophages: a tempting perspective for bladder cancer immunotherapy’
Now entitled “Tumor-associated macrophages in bladder cancer: biological role, impact on therapeutic response and perspectives for immunotherapy”
Dear Editor and Reviewers,
We thank you for the insightful comments on our review. We believe they improved our review and made it clearer and easier to read. Please find our point-by-point reply. As major changes were made, the manuscript version is without track changes to facilitate the reading. We hope that our new version complies with the standards for publication in Cancers and thank you for your kind consideration.
Reviewer #3.
The review by Leblond et al., „Targeting tumor-associated macrophages: a tempting perspective for bladder cancer immunotherapy” is very timely and provides a comprehensive overview about macrophages in bladder cancer, link tumor-associated macrophages to clinical outcomes as well as how macrophages are affected by different anti-cancer treatment regimens. The figures and tables are exceptionally well designed.
We thank the reviewer for the nice comments.
Minor comments:
- Line 303: scRNA-seq is only briefly mentioned. It would very informative to add a whole paragraph to this chapter and provide more insight into the diversity/complexity of TAMs in bladder cancer as revealed by scRNA-seq. Several other scRNA-seq papers could be included, e.g. https://www.biorxiv.org/content/10.1101/2020.07.09.195685v1.full. How do bladder-resident macrophages compare to TAMs in bladder cancer? Are TAMs in bladder cancer different to TAMs in other cancers? (e.g. does one find SPP1, C1QC TAMs also in bladder cancer?).
We thank the reviewer for this comment. While we aren't confident citing a pre-print manuscript, we agree with the importance of TAM diversity/complexity in bladder tumors. We therefore emphasized the results obtained by the scRNA-seq of TAM in the paper of Wang et al., clinical cancer research, 2021. This is now found in the line 152 of our revised manuscript.
It seems that chapter 3.3 is missing or chapter 3.4 needs to be re-labeled to 3.3.
We thank the reviewer for this remark. It was a mistake in the section labelling. We changed that (the section 3 is now the section 2 in our revised version).
Round 2
Reviewer 1 Report
The authors have corrected the manuscript properly.
Reviewer 2 Report
Authors provided an updated version of the manuscript addressing the required improvements. I suggest to consider te manuscript for publication.